# A machine learning driven computationally efficient horse shoe shaped antenna design for internet of medical things

Umhara Rasool Khan[1], Javaid A. Sheikh[1]*, Aqib Junaid[2], Shazia Ashraf[1], Altaf A. Balkhi[1]

1 Department of Electronics & IT, University of Kashmir, Srinagar, Jammu & Kashmir, India, 2 School of Management, State University of New York at Buffalo, Buffalo, New York, United States of America

* sheikhjavaid@uok.edu.in

**Data Availability Statement:** Data underline the results presented in this manuscript shall be publicly available from the date of publication.

## Abstract

With bio-medical wearables becoming an essential part of Internet of Medical things (IoMT) for monitoring the health of workers, patients and others in different environments, antenna play a pivotal role in such wearables. In this communication, a novel Horse shoe shaped antenna (HSPA) meant for such wearables is presented. The vitals of the workers, patients etc. are collected and sent to the IoMT platform for ensuring their safety and monitoring their physical wellbeing. In this article, regression-based Machine learning (ML) techniques are used to facilitate the design of Horse shoe shaped patch antenna to predict the frequency of operation, radiation efficiency and Specific Absorption Rate (SAR) values to accelerate its design process for on-body applications. The HSPA designed resonates at 2.45 GHz in the frequency band of 1.75–2.98 GHz with SAR of 1.89 W/kg for an input power of 16.98 dBm, peak gain of 1.91 dBi and radiation efficiency of 62.07% when mounted on the human body. 1080 samples of data comprising of three EM parameters have been generated using a conventional EM tool by varying the physical and electrical parameters of the design. A detailed comparison of the five regression-based ML algorithms is presented, and it is observed that the ML models help in efficient use of resources while designing an antenna for bio-medical applications.

## Section I. Introduction

Currently, we are living in a digital era where every engineering problem needs a quick and cost-effective solution. To a researcher, the idea of optimization in the Electromagnetic (EM) field seems to be a strenuous task with tedious time-consuming EM simulation for calculating the optimal parameters of the antennas/microwave components for obtaining the desired EM response. There have been serious efforts to address the challenge of optimization under the given constraints for better performance of the system and one such effort is employing Machine learning-based approaches to address the complex and multifaceted challenges while designing the system [1]. Once this approach is applied efficiently, there is no need for repetitive time-consuming EM simulation. In ML techniques, a hidden relationship between the

**Funding:** It is submitted that University Grants Commission Govt. of India under Junior Research Fellowship scheme financially supports the study. It is submitted that none of the authors receives salary from the funder other than the fellowship receives by the author Mrs. Umhara Rasool. The authors received no specific funding for this work other than the fellowship by one of the authors namely Mrs. Umhara Rasool.

**Competing interests:** The authors have declared that no competing interests exist.

input and output parameters of the design is determined and accordingly based on this relationship model, future predictions or decisions are made [2].

As cutting-edge disclosures are within rise in wireless communication systems, a number of complex antenna design are being proposed. Several ML techniques have been discussed in the existing literature for increasing the optimum usage of resources (time, human and computational resources) while designing and optimizing the geometrical parameters of antenna for a defined EM response. Artificial Neural Network is an effective tool that can accelerate the optimization process [3, 4]. An inverse ANN model is proposed in [5] to address the multi-objective problem with a large number of geometrical variables for antenna modeling where the performance parameters are set as the input for the model and the corresponding geometrical variables are the output. A fast and accurate tool known as Support Vector Machine (SVM) has been used in [6] for designing a rectangular array and predicting its EM behavior. In [7], a combined neural network and pole-residue-based transfer function model is designed for parametric modelling of microwave components to predict their EM response. An efficient Gaussian Process Regression technique for modeling of microstrip patch antenna is proposed in [8] to predict its physical or electrical parameters. In [9–12], ML has been used for predicting and optimization of various fractal inspired antennas by employing particle swarm optimization (PSO) and firefly optimization with ANN. A DNN is employed for determining the resonant frequency of E shaped microstrip antenna in [13]. Different regression-based ML models are used to predict design parameters of Slotted Waveguide Antenna for the desired frequency of operation and specified sidelobe level ratios (SLR) in [14–18]. In [19] the effect of design parameters affected by epistemic uncertainty on the performance of textile-based antenna is reported. Various performance parameters like gain, radiation efficiency, resonant frequency and directivity of microstrip slotted patch antenna (free space) are predicted using SVM in [20, 21]. A comparison between the ANN and SVM model developed for the reported design is carried out in the above-mentioned work.

Bio-medical wearables form an important component of Industrial Internet of things (IIoT) and Internet of Medical things (IoMT) where vital signs of workers and patients are continuously collected and send to the IIoT/ IoMT platforms for detecting any sign of fatigue or other health issues. This can ensure the safety of the employees which in turn increase the productivity, efficiency and gains of the business and at the same time ensure wellbeing of the patients thereby reducing the load on health institutions. At present, control, safety and analysis are the main driving factors for the development of Wearable Industrial Internet of things (WIIoT) and IoMT. The market of WIIoT is expected to grow from $1.1 billion in 2019 to whopping $1.86 billion by 2024 as per 2020 Markets to Markets Report. In these bio-medical wearables, antenna is an indispensable part of the system that helps in sending and collection of bio-medical data.

So, keeping in view the above discussion, an antenna meant for monitoring of the vitals is designed with the help of the conventional EM tool in conjunction with the ML tools. In this study, ML is used to facilitate in designing of the antenna for biomedical applications keeping in view the need for optimal radiation efficiency and SAR within the prescribed limit on mounting of antenna on a human body. It needs to be mentioned here that antennas designed for on-body application is a challenging task as the RF waves interact with different tissues of human body making the impedance matching a difficult task. Also, there is significant reduction in gain of the antenna meant for bio-medical applications owing to the gross disparity between dielectric properties of air and tissues of human body and change in the frequency of operation for which it is designed. Therefore, it becomes imperative for the RF engineer to design a wideband, flexible, biocompatible and miniaturized antenna for biomedical applications [22–23].

A compact hybrid Moore's fractal inspired antenna has been proposed recently where an attempt is made to reduce the SAR within the prescribe limits for maximum input power of 24 dBi [24]. A novel circularly polarized antenna backed by artificial magnetic conductor is presented in [25] for Wireless body area network (WBAN) applications. The reported antenna in [25] achieves peak gain of 7.4dBi along with the radiation efficiency of 97.4% and SAR of 0.0264 W/kg. A wearable belt antenna made of textile material backed by (Electromagnetic Band Gap) EBG backed ground is proposed for smart On-body applications in [26]. A better antenna (when mounted on human body) in terms of SAR, radiation efficiency and gain has been proposed in the above reported work. A flexible tri-band antenna meant for wearable devices and body-centric wireless communications operating at the key frequency bands is proposed in [27]. Rogers RT 5880 is used as a substrate with a mere thickness of 0.254 mm and an overall geometrical dimension of $15 \times 20 \times 0.254$ mm$^3$. This inventive design features a truncated corner monopole accompanied by branched stubs fed by a coplanar waveguide. The stubs, varying in length, serve as quarter-wavelength monopoles, facilitating multi-band functionality at 2.45, 3.5, and 5.8 GHz. A triband quarter-wave monopole antenna with potential for ISM, IoT and sub 6 GHz applications is presented in [28]. The proposed antenna offers a radiation efficiency of more than 75% in all the three bands and gain varies from 1.84–2.72 dBi. However, the performance of the antenna in presence of human body hasn't been evaluated in the study.

After understanding the challenges in antenna design meant for On-body applications, it is concluded that the Machine Learning shall provide a way out for overcoming these challenges. It is a worthwhile point to note down that the simulation of an antenna over the human tissues requires not only high end systems but also requires a lot of other resources (time, human and computational resources) in comparison to the free space-based simulations. Based on these observations, it is concluded that ML shall be used to predict the desired behavior of antenna in terms of SAR, frequency of operation and radiation efficiency without carrying out strenuous EM based simulation using only conventional tools. Therefore, after using ML tools a researcher can decide with less usage of resources, the geometrical and physical parameters of antenna and the optimum distance above which antenna needs to be placed to get an optimal antenna with desired radiation efficiency, resonant frequency and SAR. Furthermore, after having a look on the results obtained from ML tools, a researcher shall have an in-depth knowhow about the input and output parameters of the design. So, it is concluded that the conventional EM tools need to be operated in conjunction with ML tools for optimum usage of resources.

In this work, an antenna operating in the ISM band is designed initially using a conventional EM tool (HFSS). Thereafter, a comprehensive comparative study of different regressive ML algorithms is performed to estimate the operating frequency, radiation efficiency and SAR of the proposed antenna in the human environment. The different ML algorithms employed in this study include SVR, DT, RF, ANN and Gaussian regression analyses. This study shall form an important foundation for the future of Machine learning in bio-medical applications.

This work was completed in the following phased manner:

**Step 1.** A data set of physical and electrical parameters of the design is developed for the desired EM response (frequency of operation, radiation efficiency and SAR). A dataset of 1080 was generated by simulating the horse shoe shaped antenna using the HFSS software.

**Step 2.** The dataset generated in the first step is used to train the above-mentioned ML algorithms in a supervised manner. The ML models so developed can be used to predict the desired responses for given physical and geometrical parameters of the design.

**Step 3.** The performance of the ML models developed in the second step is evaluated using different metrics. Based on these metrics, a thorough comparison is performed between the developed models.

**Step 4.** Comparison of the results is done between the EM responses achieved through the best ML model developed in this study and a conventional EM simulator.

This write up is divided into five sections; the introduction section gives us a deep insight about the problems faced by RF engineers while using EM tools in a standalone manner for different EM problems and also discusses about the recent solutions provided by researchers for these problems, proposed design section discusses the methodology used for designing an optimized antenna for bio-medical applications; challenges that are faced while utilizing ML approach for EM problems are discussed in Challenges employing ML techniques Section. In proposed ML approaches for antenna design section, ML models and the statistical metrics used thereof used in this study are briefly discussed whereas the details of the training of the dataset generated using the conventional EM tool is discussed in training and validation Section. The results obtained by using EM and ML tools are evaluated in results and discussion section. The write up is concluded in the conclusion and future section with the concluding remarks in addition to the future work that needs to be carried forward.

## Section II. Proposed design

An antenna fabricated on ROGERS RT/DUROID 5870 for bio-medical applications is proposed in this work. A schematic diagram of the proposed antenna is illustrated in Fig 1. The final design is achieved after undergoing a number of evolutionary stages. The prospective antenna consists of a top radiator with horizontal horse shoe shaped slots and symmetrical open-ended rectangular slots at the top edge.

In order to verify the model proposed, the optimized dimension with regard to the application is selected and fabricated on ROGERS RT/DUROID 5870 with dimensions 34 mm x 28 mm (Dielectric Constant = 2.2, Dielectric tan δ = 0.0012 Thickness = 1 mm). The prototype fabricated has a ground dimension of $21*10$ mm$^2$ with a rectangular slot embedded in the middle and two hybrid U-shaped cuts at the top edge of the ground plane. The optimized dimension of the middle slot and hybrid U-shaped cuts are illustrated in Fig 1. The antenna designed is simulated on a multilayer human phantom in HFSS to record EM responses. The multilayer human phantom designed in this work consists of four tissues; skin, fat, muscle and bone as reported in [20].

The modification in the dimensions of the ground in the proposed design has been carried with an aim to select the desired band of operation along with the better impedance matching. This concept is supported by the fact that the modification in the dimensions of the ground leads to change in the impedance of antenna and at the same time helps in selecting the desired frequency range of operation[15]. The dimensions of the prototype designed is 0.272 $\lambda_0 \times 0.224\lambda_0$ where $\lambda_0$ is the wavelength of the lowest resonating frequency. A rectangular feedline of length $L_f$ and width $W_f$ is used to feed the top radiator. All the optimized geometrical parameters of the design are tabulated in the Table 1.

## Section III. Challenges in employing ML techniques in complex electromagnetic (EM) problems

It is a well-established fact that with the advent of Machine learning, the field of EM has revolutionized. However, at the same time adopting of ML approach in EM problems poses a lot of

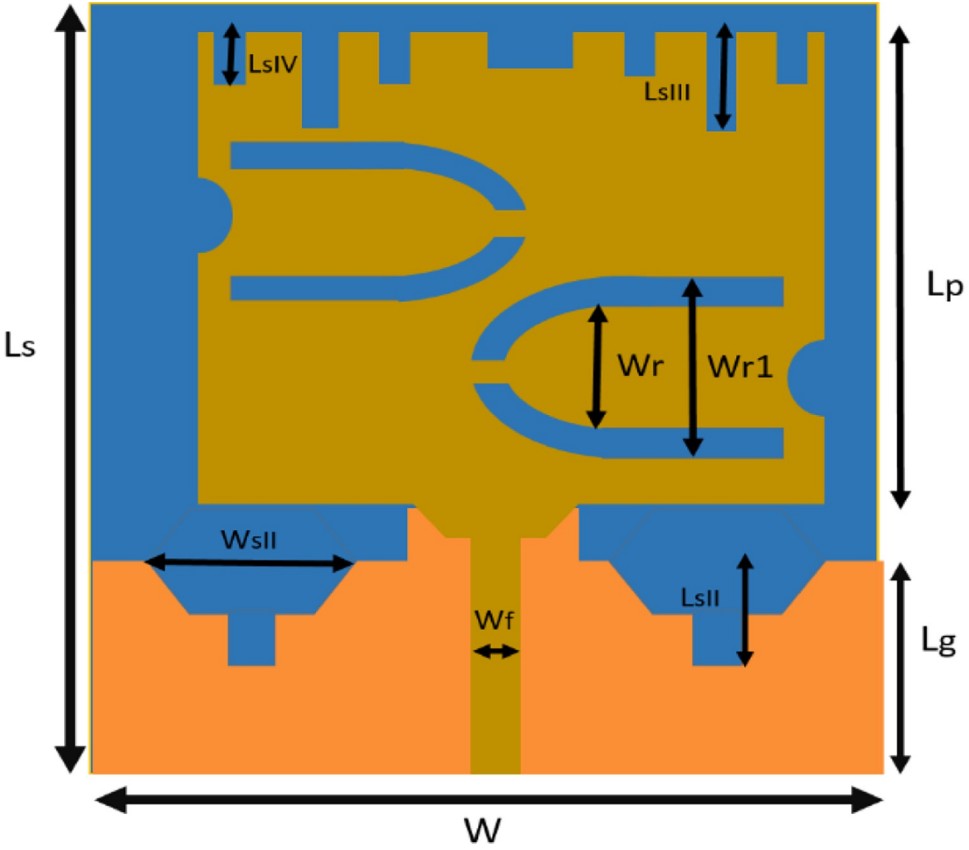

**Fig 1. Geometrical layout the antenna.**

limitations. The most common challenges while utilizing ML in EM problems are discussed as follows:

1) Identification of the Problem and thereof Problem Formulation: Starting with incorrect assumptions frequently results in worthless time-consuming results. So, it is better to know beforehand which aspect of the issue is best to focus on

2) Selection of learning algorithm: Since there are a large number of ML algorithms available, so it is always very difficult to decide which algorithm to employ in our problem from such a large number of algorithms. So before selecting any algorithm, it needs to be observed as to what is being predicted and also the type of data at our disposal. Therefore, it is recommended to visualize the data before choosing the algorithm.

**Table 1. Parameters of the geometrical layout of the proposed design.**

| Parameter | Value (mm) | Parameter | Value (mm) |
|---|---|---|---|
| $L_S$ | 34 | $L_{SII}$ | 7.12 |
| W | 28 | $W_R$ | 5 |
| $L_g$ | 13 | $W_{RI}$ | 9 |
| $W_f$ | 2.9 | $L_{SIII}$ | 2 |
| $L_f$ | 14 | $L_{SIV}$ | 1.5 |
| $L_{sII}$ | 9 | D | 5 |
| $L_P$ | 16 | $W_P$ | 24 |

3) Getting enough and reliable data: In EM designs large number simulations are needed to obtain enough data for efficiently training the model so as to minimize the error percentage in the predicted data

4) Data Pre-processing: Multiple preparation steps such as cleaning of data, normalization and feature selection are performed on the data before it is fed to model for ensuring that the algorithm performs efficiently.

5) Debugging the algorithm: It can be difficult to determine what to do next when problems of high variance and bias arise. To determine the necessary steps, it is essential to apply diagnosis techniques like plotting the learning curves. Moreover, the model's performance can be enhanced by using techniques like regularization, feature selection, and tuning the hyper parameters.

## Section IV. Proposed ML approaches for the design of antenna for biomedical applications

To begin with different ML approaches, we need to define our input and output feature vector. In this context, let $X = [x_1, \quad x_2, \ldots, x_{N_i}]$, $Y = [y_1, \quad y_2, \ldots, y_{N_j}]$ be the input and output feature vector respectively where $N_i$ represents the number of features in the input vector and $N_j$ represents the number of features in the output vector. Here, our input vector consists of $N_i = 5$ features that include relative permittivity of the substrate ($\epsilon_r$), slot length of the rectangular slot etched in the ground ($L_{sI}$), radii of the circular slot etched in the radiating patch (p = D/2), length of the ground plane ($L_g$) and height of the antenna from the human body ($H_b$).

In this study, we have employed number of regression-based machine learning algorithms to discover the non-linear relationships between X and Y which in turn help us to determine the desired EM responses for the given physical and electrical parameters. The mathematical description of the ML techniques used in this work is discussed in the below subsections wherein the hyper parameters used are also specified.

### a. Decision tree

Decision tree (DT) alternatively known by the name of predictive model is a supervised ML algorithm (created in the form of a tree) that can be used to solve both the regression and classification tasks [16]. DTs are decision support tools that use tree like model of decision with tree structure based on if-then-else questions. The DT algorithm works by continuously splitting larger data set into smaller subsets and thereof an associated decision tree develops at the same time. In order to ensure better performance of the complex problems, a lot of data should be used for training of the DTR model. So, the size of the dataset should determine the size of the tree and the number of nodes in it.

### b. Random forest

Random forest is a subset of ensemble learning method which combines the results of multiple de-correlated decision trees to reduce variance. A bootstrap sample of the training dataset is drawn, and a subset of the initial features is selected. Each decision tree is then fitted to the sample, and the results are averaged to make a prediction. In the random forest model, the predicted value $\hat{y}$ for an input sample x is computed by the given mathematical expression:

$$\hat{y} = \frac{1}{N} \sum_{i=1}^{N} T_i(x) \tag{1}$$

where T (.) represents a decision tree and N represents the total number of decision trees which was chosen to be 30 in this model with maximum depth of 5.

### c. Support vector regression

SVR being the most frequently used type of support vector machine in regression analysis is motivated by the ubiquitous Support vector Machines (SVMs) classifier. The basic idea in Support Vector Machines (SVM) is to find an optimal hyperplane that best separates the data into two classes and then use this hyperplane to determine the regression function [14]. SVR is characterized by the use of kernel and the margin control. Since the function to be predicted is non-linear, kernels are used to map the input into a higher dimensional space known as the kernel space [15]. The loss function of SVR is quantified by the expression:

$$\mathcal{L}_{SVR}(w, b) = \frac{1}{2}|w|^2 + C\sum_{i=1}^{m} |w.\phi(x_i) + b - y_i|_\varepsilon \tag{2}$$

Where $\phi(.)$ is the transformation that maps inputs from the attribute space to the kernel space, C is a regularization term, and $|\ |_\varepsilon$ is the $\varepsilon$-sensitive loss. The optimal values of $\varepsilon$ and C selected in this work are 0.001 and 50 respectively.

### d. Gaussian regression

Gaussian Process Regression is a type of machine learning that uses Bayesian Probability theory to fit the data points. It does so by defining a prior probability distribution over functions and then uses the data to update that distribution. This prior distribution is often chosen to be a Gaussian process, which allow us to interpolate between the data points and is closely related to KRR and linear regression with radial basis function.

The loss function of a GPR model can be expressed mathematically as:

$$\mathcal{L}_{GPR}(\theta) = \frac{1}{N} * \sum [(y_i - f(x_i, \theta))^2] \tag{3}$$

where N represents the number of points in the dataset, $\theta$ is the set of hyper parameters for the model $y_i$ is the observed value of the target variable for the i[th] data point, $x_i$ is the i[th] data point and $f(x_i, \theta)$ is the predicted value of the target variable for i[th] data point.

### e. Artificial neural network

Artificial neural network (ANN) being one of the artificial intelligence techniques is based upon the concept of imitating the working of brain wherein information travels through neurons between organs and the brain. In simpler words, NN can be considered as a computational black box that imitates any system having an input-output relationship through multiple layers of neurons. In this system, there is parallel processing of the information which enables the NN to learn the principle that the coefficients of connections between neurons need to be updated by minimization of error. A NN model comprises of three types of layers which includes input, hidden, and output layers. Moreover, each layer in the model has neurons that are connected utilizing their weight factors. In fully connected ANN, the regular feed-forward operation is mathematically represented by

$$\begin{aligned} h_i^{x+1} &= w_i^{x+1} a^x + b_i^{x+1} \\ a^x &= f(h_i^{x+1}) \end{aligned} \tag{4}$$

where a represents the output vector from a layer, x is the layer index, i represents the unit

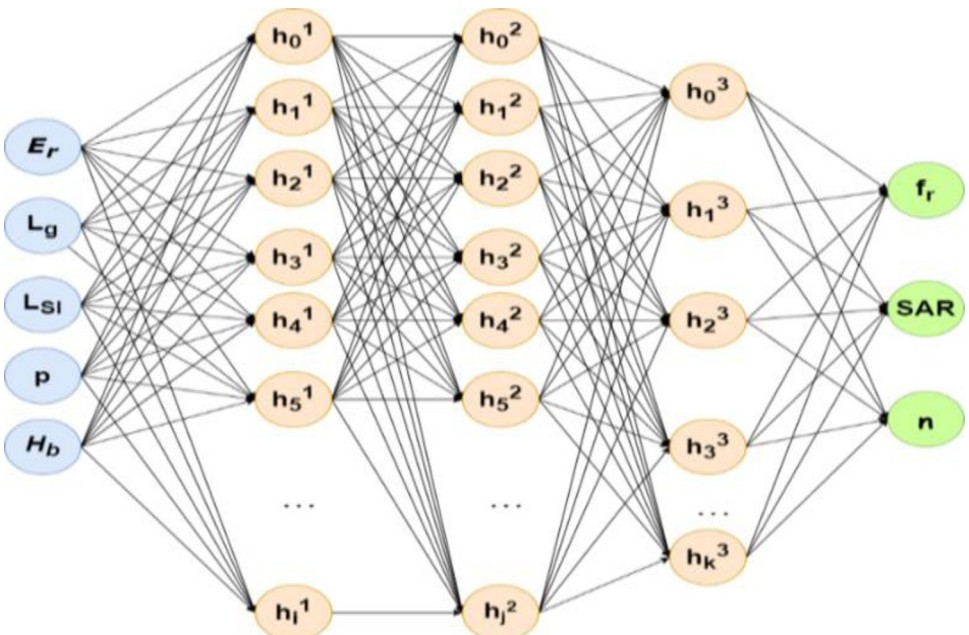

**Fig 2. Architecture of ANN model used in this work.**

index, h is the input vector into a layer, w is the vector representing the weights, b is the bias parameter vector, and $f$ is the activation function. The network is trained to minimize the mean squared error with loss function given by:

$$\mathcal{L}_{ANN} = \frac{1}{m}(\sum_{i=1}^{m}(\hat{y}_i - y_i)) \tag{5}$$

The number of inputs and outputs typically determine the number of neurons in the input and output layers respectively. The number of neurons used in the hidden layer depends upon the complexity of the problem, however too few can limit the learning capacity of the model [18].

Fig 2 illustrates the architecture of the ANN used for design optimization of the prospective antenna. Hyper parameters utilized in the ANN model include three hidden layers with 500,500 and 150 neurons in each layer respectively, ReLU as activation function, random seed of 40 and ADAM algorithm for optimization.

## f. Statistical metrics used for evaluating the performance of ML tools

Performance of the regression models in this article are evaluated using the three statistical metrics which includes; Mean absolute error (MAE), Mean square error (MSE) and the coefficient of determination ($R^2$ score).

The MAE provides a measure of the average magnitude of the differences between the actual and predicted values in the dataset. It is calculated by taking the average of the absolute values of the residuals (the differences between the actual and predicted values).

$$MAE = \frac{1}{N}\sum_{i=1}^{n}|y_i - \hat{y}| \tag{6}$$

Where $y_i$ is the desired output which is the actual value in the test set, $\hat{y}$ is the predicted output and N is the number of samples in the test set.

The second metric used to evaluate the models is MSE which can be defined as the average of the squared difference between the actual and the predicted values. It measures the spread of the residuals and can be expressed as:

$$MSE = \frac{1}{N} \sum\nolimits_{i=1}^{n} (y_i - \hat{y})^2 \tag{7}$$

The third metric used for evaluating the performance of the regression models in this article is $R^2$ score. In regression models, it represents the proportion of the variance for a dependent variable that is explained by an independent variable or variables and ranges from 0 to 1. It can be calculated by the following equation:

$$R^2 = 1 - \frac{\sum_{i=1}^{n} (y_i - \hat{y})^2}{\sum_{i=1}^{n} (y_i - \bar{y})^2} \tag{8}$$

Where $\bar{y}$ is the mean of the actual values in the test set.

## Section V. Training and validation

A dataset of 1080 samples was obtained after simulating Horse shoe shaped antenna design meant for biomedical applications using HFSS EM simulator. Each individual sample consists of different design and electrical parameters as input parameters of the proposed antenna optimized for achieving the desired EM response (resonant frequency, SAR and gain). The dataset was partitioned in the ratio of 0.7:0.3 for training, validation and testing respectively. All the operations were performed on Intel Core i7-7700 3.6 GHz processor with 8 GB RAM. Moreover, The GPR, RF, SVR, and DT ML models were developed and trained using the Scikit Library whereas the ANN model for this work was developed using the Tensorflow library and trained using ADAM algorithm [12, 16].

The details of the dataset generated for the input vector consisting of relative permittivity of the substrate ($\epsilon_r$), length of the rectangular slot etched in the ground ($L_{sI}$), radii of the circular slot etched in the radiating patch (p = D/2), length of the ground plane ($L_g$) and height above which the antenna is placed from the human body ($H_b$) is illustrated in Fig 3.

In this study a 6-fold cross-validation method was used to substantiate the validity of the methods utilized in this paper. The cross-validation process works in a manner whereby the data set is divided into K equal and repeated K times. However, it needs to be kept in consideration that only and only once each subset must be used for ensuring that all of the data

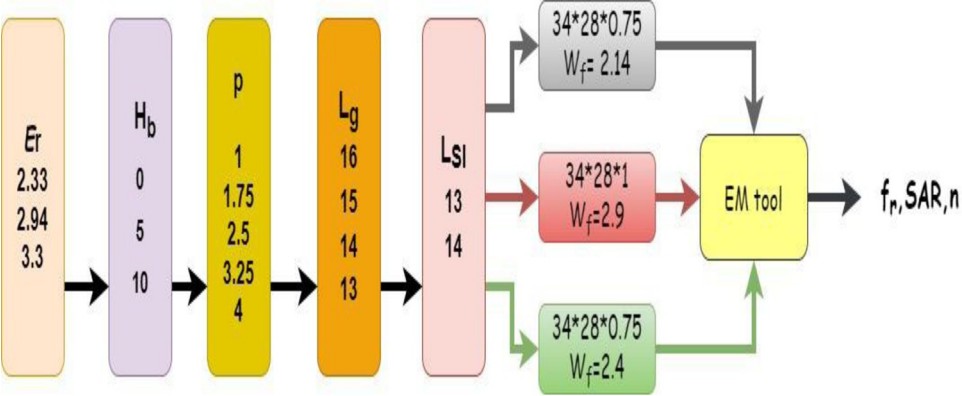

**Fig 3. Dataset generation for training of regression based ML models.**

generated is used for both training and validation. Fig 3 illustrates the generation of dataset. The full data set was divided into 6 subsets. For each iteration, 756 data points were used for training and 324 were used for testing and validation.

## Section VI. Results and discussion

### a. Conventional EM tools used in antenna design

The optimized HSPA was designed using both the conventional EM tool along with the ML tool and thereof the final prototype was consequently fabricated on RT/Duroid 5880. The radiation pattern and return loss measurements of the prototype were performed at the University of Kashmir, SPACE lab. The fabricated prototype and measurement setup is illustrated in Fig 4.

From Figs 5 and 6, it is evident that HSPA resonates at 2.67 GHz in free space whereas it resonates at 2.47 GHz and 2.98 GHz when placed on the body. The prototype works in the frequency band of 2.19– 3GHz and 2.32–3.04 GHz.

The difference between the measured results in free space and body can be attributed to the detuning effect of human body. Furthermore, it is demonstrated in Fig 5 that the measured bandwidth is less than the simulated bandwidth in addition to the slight shift of resonant frequency towards the higher value in the measured result.

This variation observed between the simulated and measure results occur due to small geometrical inaccuracies that happen during fabrication and measurement assembly.

In order to understand the effect of human tissues on the performance of antenna, the prototype is placed at different positions above the tissues. This effect on reflection coefficient is

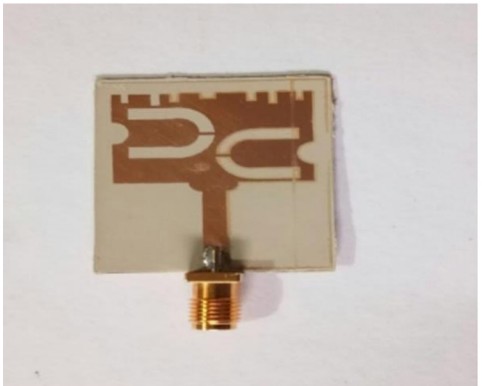
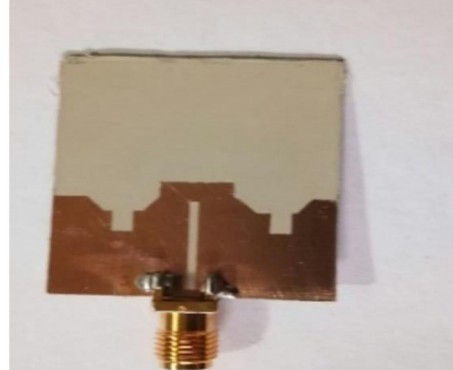
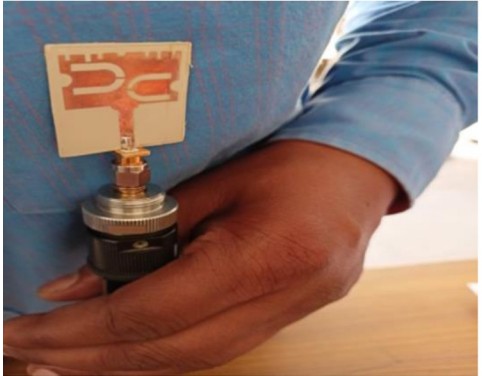
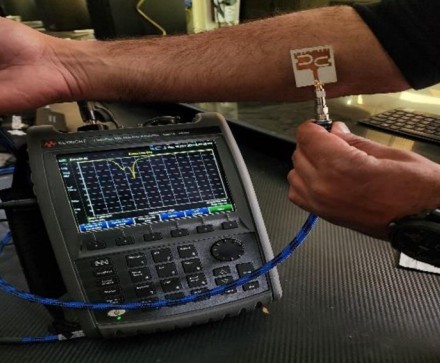

**Fig 4. Fabricated prototype under measurement.**

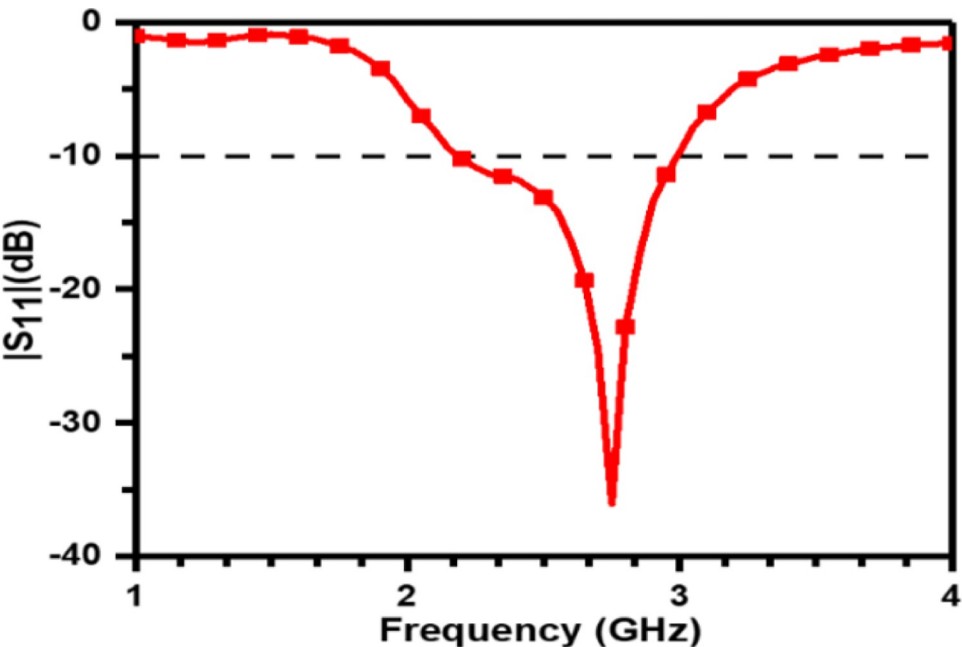

**Fig 5. Measured reflection coefficient in free space.**

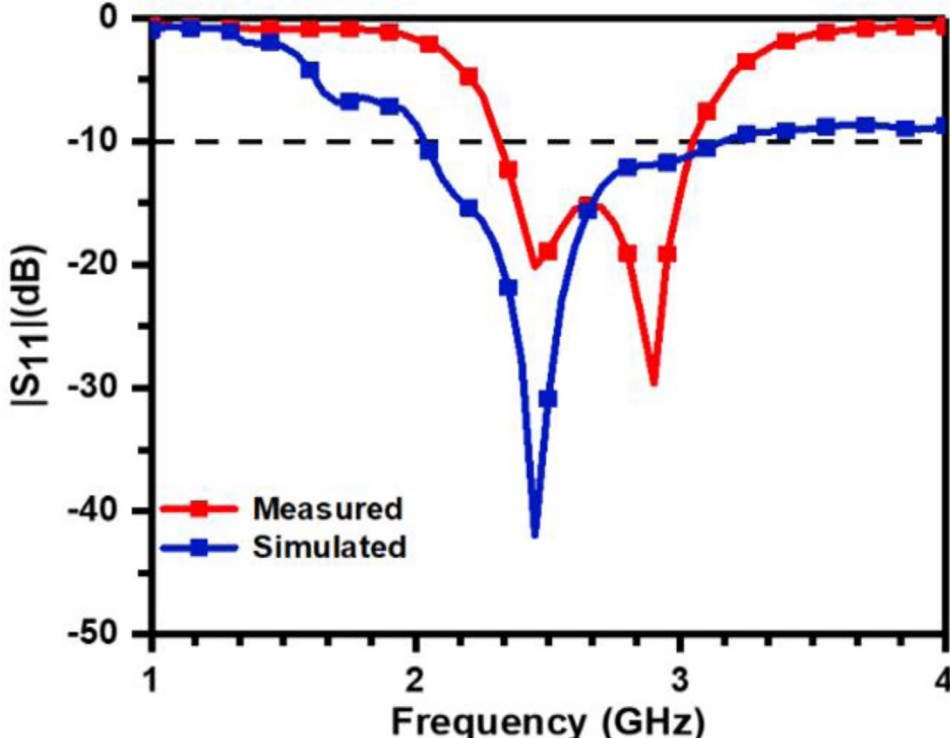

**Fig 6. Simulated and measured reflection coefficient plot on human body.**

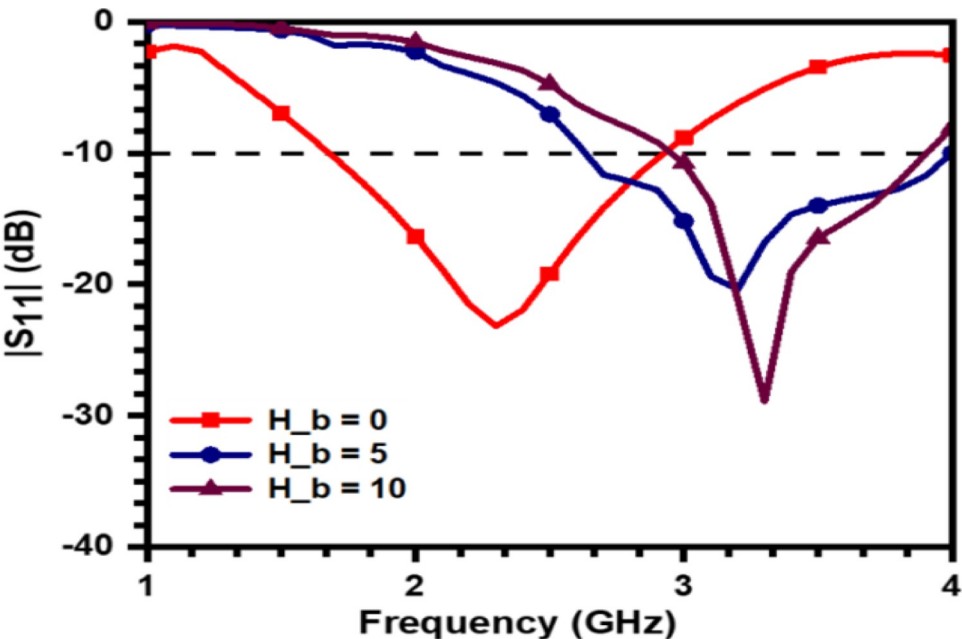

**Fig 7.** Simulated reflection coefficient plot for different values of $H_b$.

summarized in Fig 7, wherein the resonant frequency and the band of operation gets shifted towards higher frequencies as the height of the prototype is increased resulting in more separation between the body and antenna. Fig 8 gives us insight into the Poynting vector of the design that actually defines how the energy is transferred.

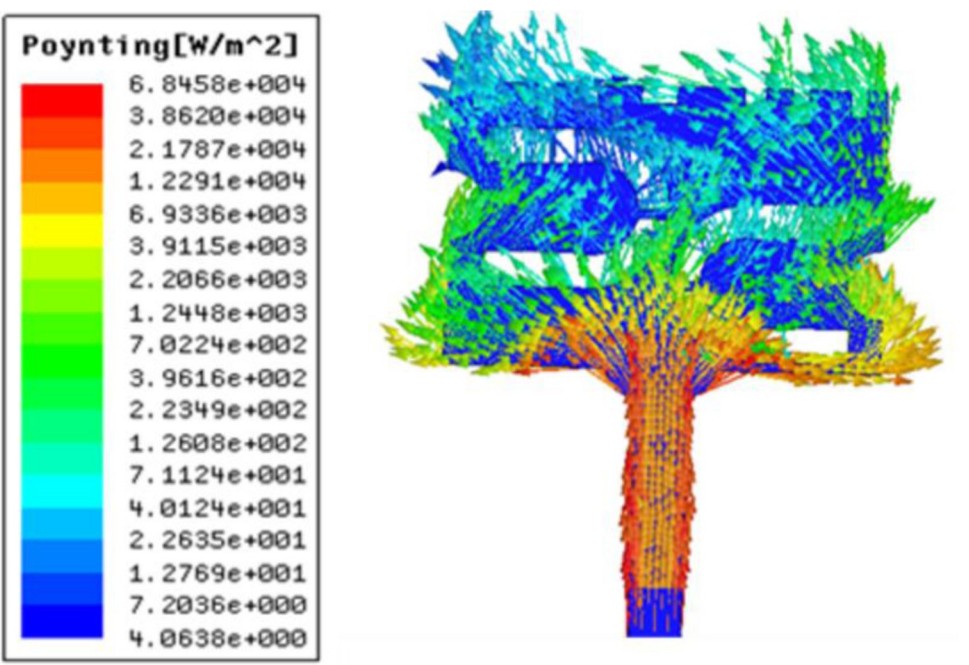

**Fig 8. Plot of poynting vector on radiating patch.**

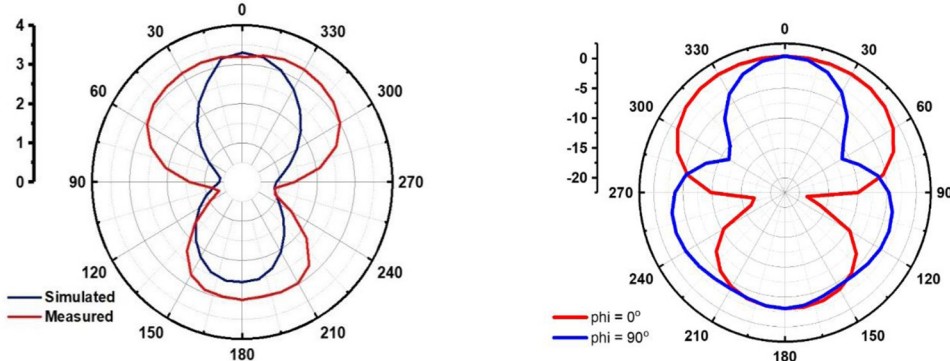

**Fig 9.** a) Simulated and measured radiation pattern for $\phi = 0°$ in free space b) Simulated radiation pattern of design when mounted on body. The Fig 10 shows the Polar plot of the proposed antenna design and it is observed that directionality is maintained as required.

Fig 9 A) and 9 B) illustrates the simulated and measured radiation pattern of the prospective design at $\phi = 0°$ in free space and simulated radiation pattern when mounted on human body respectively.

Since the prototype needs to be mounted on the human body, SAR of the design is evaluated. It is observed from Fig 11 that for an input power of 750 mW, the average SAR of 1.01 W/kg (over 10 g tissue) is obtained. Figs 12 and 13 illustrates the variation of electric field and surface current distribution on the radiating patch respectively.

## b. Regression based ML models in antenna design

In this work an attempt is made to predict frequency of operation, radiation efficiency and SAR of the antenna using few regression-based ML models so that the RF engineers don't rely

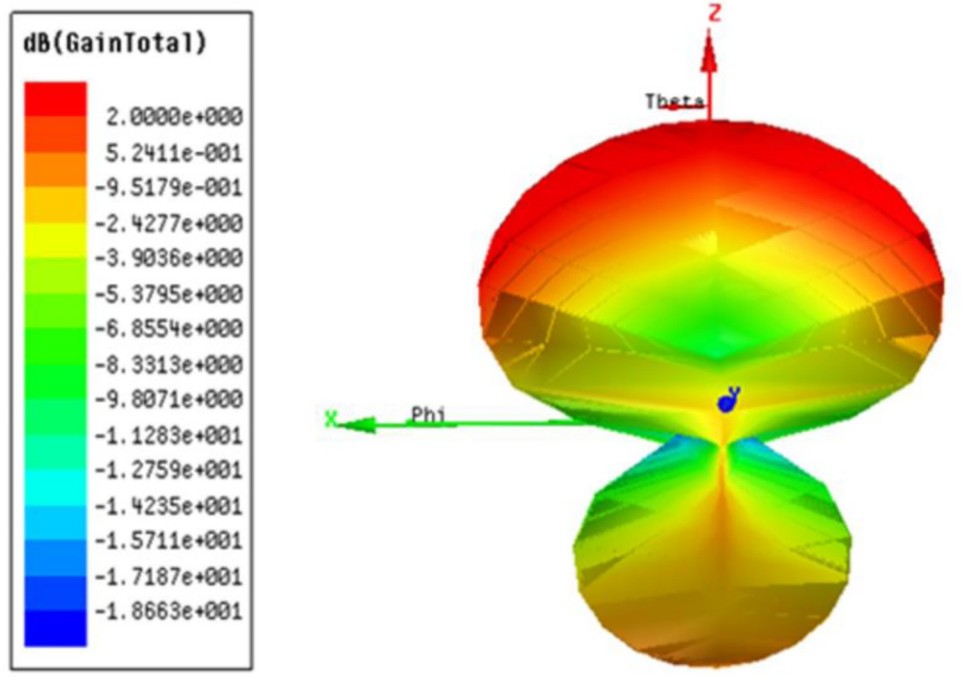

**Fig 10. Polar plot of the antenna design.**

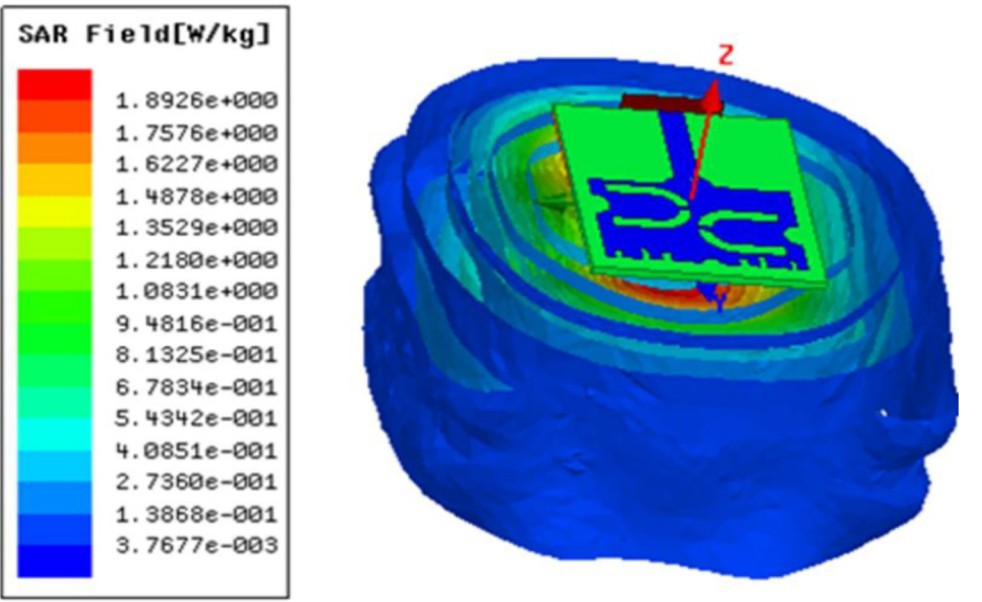

**Fig 11. Specific absorption rate of the prototype.**

in such circumstances on tedious simulation-based optimization process on standalone EM simulators.

To evaluate the performance of the trained ML models, a 6-fold cross-validation is performed and the statistical summary of the three metrics is illustrated in the boxplots below. Furthermore, performance of the ML models is summarized in Table 2.

On analyzing Fig 14, it can be observed that the DT achieves best cross validation results with a highest $R^2$ (median) score of 0.99 and a small IQR of 0.984 to 0.986 followed by RF and

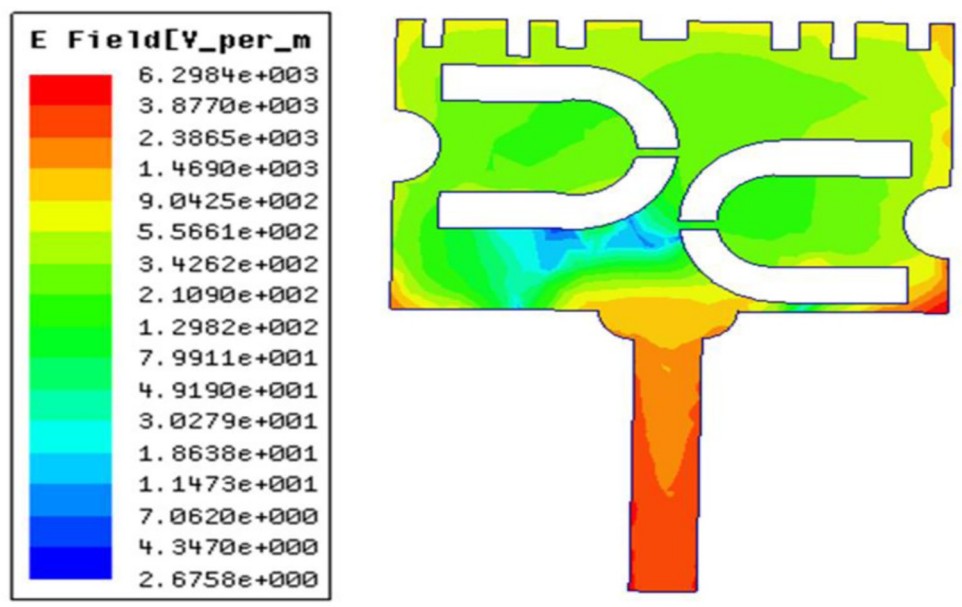

**Fig 12. Electric field distribution on the radiating patch.**

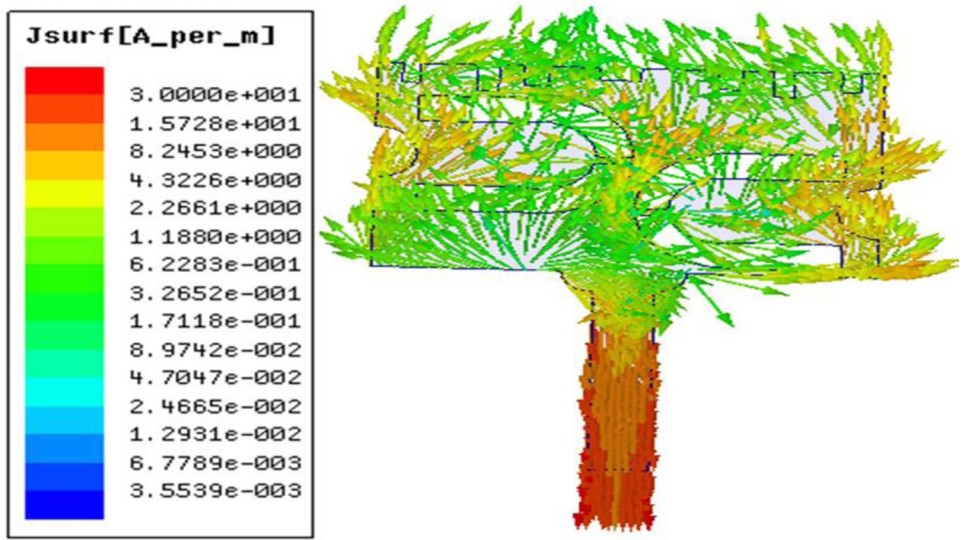

**Fig 13. Surface current distribution on the radiating patch.**

GPR with $R^2$ score of 0.985 and 0.98 respectively. The worst performing model among these models is SVR in terms of $R^2$ score.

It is observed from the Fig 15 that among the regression-based ML models used in this work, RF outperforms the other four models with a median MSE of 0.001911 and IQR of 0.0002 to 0.0025. RF is followed by DT and GPR.

Similarly, in Fig 16, DT has a lowest MAE median of 0.04 followed by RF, GPR, ANN and SVR with MAE median of 0.042, 0.045, 0.051, 0.053 respectively.

Table 3 provides us the insight about the performance of our optimized design obtained using ML assisted CEM software. It is observed that our prototype outperforms other design in terms of fractional bandwidth, gain and SAR for maximum input power of 750 mW.

For further verification of our claim, a random dataset of 100 samples (unbiased) was generated using the conventional EM tool. These samples were neither used for training or validation purposes previously. This dataset was fed to our developed ML models for testing their performance. It was observed that DT showed mean error of 0.05 over 100 random samples followed by RF, GPR, ANN and SVR that showed mean error of 0.084866, 0.085528, 0.18770 and 0.30146 respectively. This observation further reinforced our claim that we made based on our dataset as observed in Table 4.

## Section VII. Conclusion and future work

This work presents a novel design of an antenna operating in the frequency range of 2.32–3.04 GHz covering ISM band with peak gain of 3.1dBi in free space making it a potential candidate

**Table 2. Comparison between different ML approaches.**

| Parameter | DTR | GPR | SVR | RF | ANN |
|---|---|---|---|---|---|
| MSE | 0.0023 | 0.023 | 0.025 | 0.00191 | 0.026 |
| $R2$ | 0.99 | 0.98 | 0.96 | 0.985 | 0.97 |
| MAE | 0.04 | 0.045 | 0.053 | 0.042 | 0.051 |
| Prediction Time (Second) | 0.07425 | 0.16829 | 0.06387 | 0.289 | 0.767 |

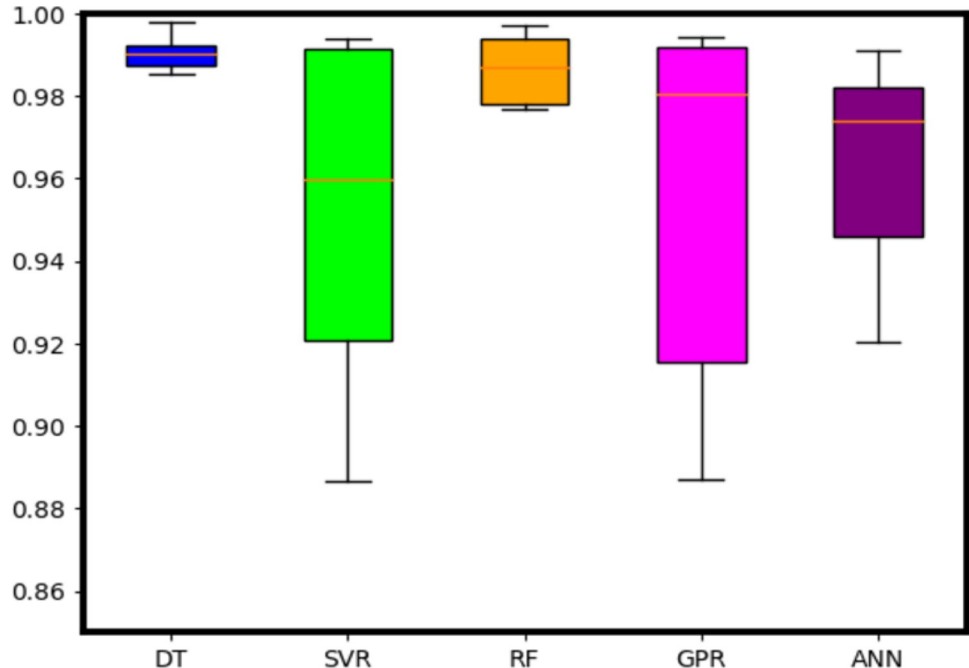

**Fig 14.** Boxplots showing the statistical metric of $R^2$ score after 6 fold cross validation.

for Wearable Industrial Internet of Things (WIIoT) and Internet of Medical Things (IoMT). Conventional EM tool are initially utilized for optimizing the physical and electrical parameters of the design and thereof 1080 samples are generated on the basis of these parameters. The dataset developed is used to train regression-based ML models: GPR, SVR, DT, RF and ANN for facilitating the optimal design based on the desired EM responses which include frequency of operation, radiation efficiency and SAR. It is observed that the DT and RF models

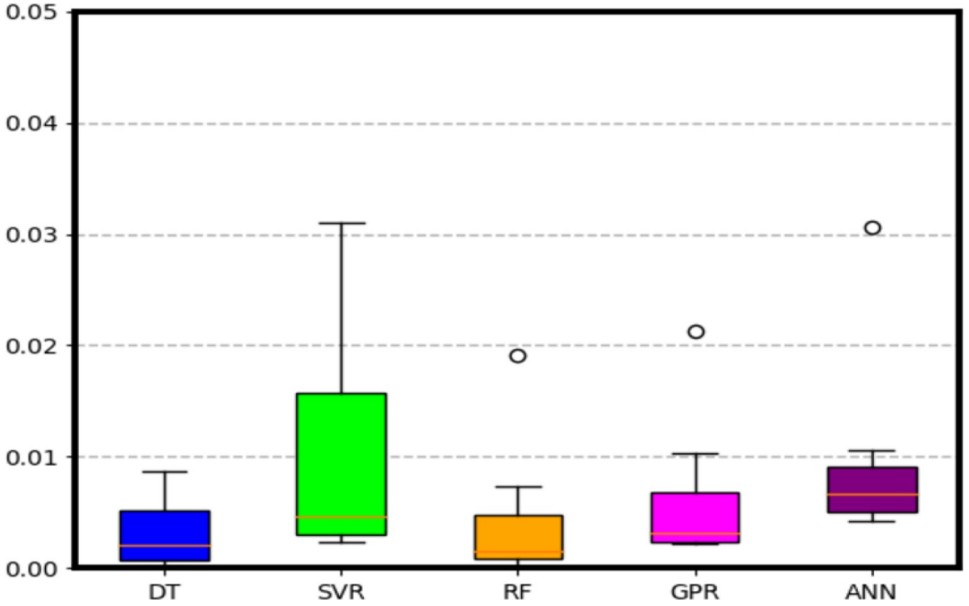

**Fig 15. Boxplots showing the statistical metric of *MSE* after 6 fold cross validation.**

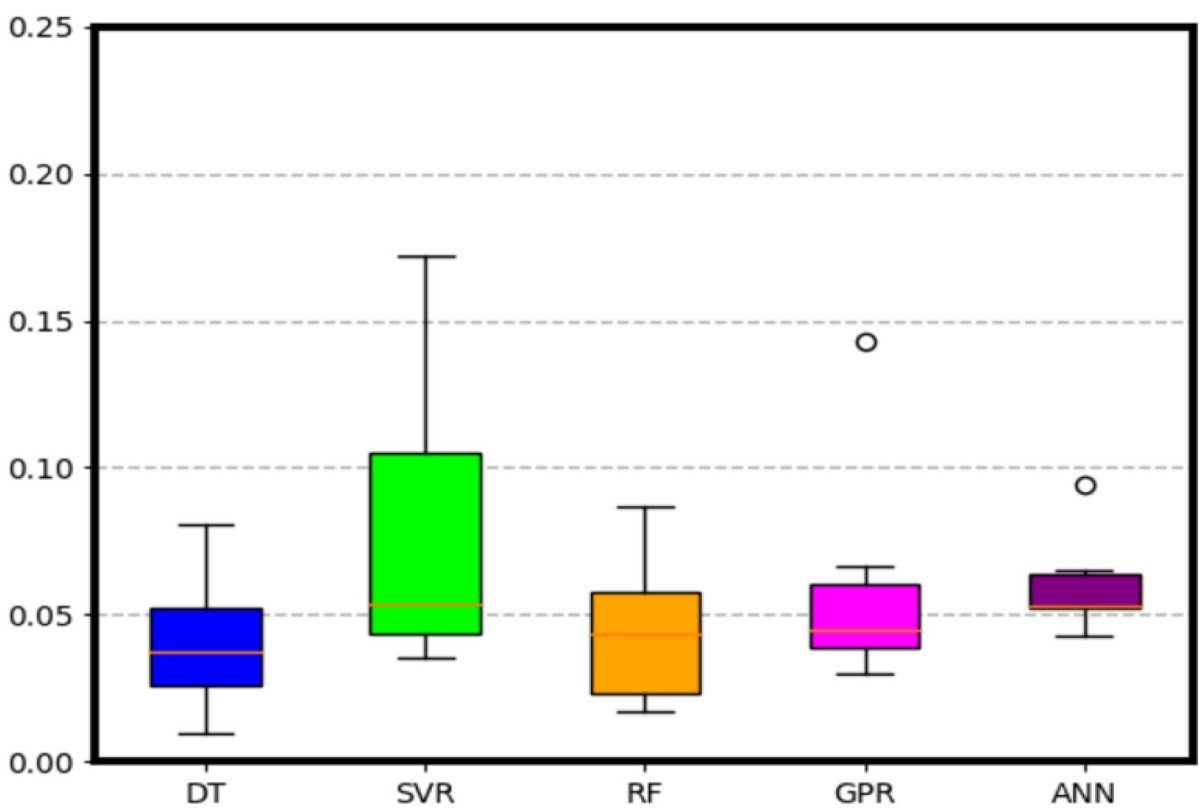

**Fig 16. Boxplots showing the statistical metric of *MAE* after 6 fold cross validation.**

outperformed the other ML models in terms of MSE, MAE and $R^2$ score. To the best of our knowledge, this is one of the few studies that uses ML tools for comprehending the effect of design parameters of antenna on SAR and radiation efficiency in presence of human body and thereof predicting the EM behavior in such scenarios. It is concluded that ML based approaches help in faster and efficient solution to complex EM problems with minimal usage of resources.

**Table 3. Comparison between the state-of-the-art prototypes.**

| Ref. | Dimensions | Resonant Frequency (GHz) | Fractional Bandwidth (%) | Gain (dBi) | Flexible | SAR(W/kg) / Input Power(dBm) |
|---|---|---|---|---|---|---|
| [19] | NA | 2.45 | 3.41 | NA | NA | NA |
| [20] | $0.503\lambda_0 \times 0.503\lambda_0$ | 2.155 | 4.6 | 7.44 | Semi-Flex | NA |
| [21] | $0.312\lambda_0 \times 0.279\lambda_0$ | 2.68/3.33/4.11 | 125.49 | NA | Rigid | NA |
| [24] | $0.135\lambda_0 \times 0.093\lambda_0$ | 1.4/3.3 | 38.5/73.7 | 2.2 | Semi-Flex | 0.02539/24 |
| [25] | $0.62\lambda_0 \times 0.62\lambda_0$ | 5.8 | 11.44 | 7.6 | Semi-Flex | 0.0264/20 |
| [26] | $1.04\lambda_0 \times 1.04\lambda_0$ | 2.45 | 30.77 | 7.94 | Semi-Flex | 0.04/26.99 |
| [27] | $0.12\lambda_0 \times 0.16\lambda_0$ | 2.45/3.5/5.8 | 5.34/7.98/38.76 | 1.08–2.99 | Semi-Flex | 1.52/NA |
| [29] | $0.318\lambda_0 \times 0.318\lambda_0$ | 2.45 | 7.75 | 2.06 | Yes | 1.95/NA |
| [30] | $0.1143\lambda_0 \times 0.2695\lambda_0$ | 2.45 | 16 | 2.52 | Yes | NA |
| [31] | $0.24\lambda_0 \times 0.145\lambda_0$ | 2.4/5.4 | 21.78/12.2 | 3.9/4.8 | Semi-Flex | 1.43/0.2 |
| **PROP.** | $0.272\lambda_0 \times 0.224\lambda_0$ | 2.45 | 51.55 | 3.1 | Semi-Flex | 1.89/0.75 |

**Table 4. Validation of test samples using different ML approaches.**

| Test Sample [$f_r$, $\eta$, SAR] | Model | $f_r$ (GHz) | $\eta$ | SAR (W/Kg) | Error | | |
|---|---|---|---|---|---|---|---|
| | | | | | $f_r$ | $\eta$ | SAR (W/Kg) |
| Sample #1 [3.3,0.9309,0.14374] | ANN | 3.297195 | 0.9567030 | 0.35847158 | 0.002805 | 0.21473158 | 0.025908 |
| | GPR | 3.10019433 | 0.9309287 | 0.3509032 | 0.1998057 | 0.20716328 | 0.000029 |
| | RF | 3.26689704 | 0.9463520 | 0.35007 | 0.03310246 | 0.20633 | 0.015452 |
| | SVR | 3.09318565 | 1.0281760 | 0.4463793 | 0.20681435 | 0.3026393 | 0.097276 |
| | DT | 3.3 | 0.927 | 0.34675 | 0.0 | 0.20301 | 0.0895 |
| Sample #2 [2.2,0.850,0.50635] | ANN | 2.16925405 | 0.88143154 | 0.4662145 | 0.03074595 | 0.04013544 | 0.076432 |
| | GPR | 2.18451231 | 0.76498162 | 0.50936154 | 0.01548769 | 0.00301154 | 0.04001838 |
| | RF | 2.22825196 | 1.181208 | 0.50715322 | 0.02825196 | 0.00080322 | 0.376208 |
| | SVR | 2.52812003 | 0.69553544 | 0.50343668 | 0.05812003 | 0.0029332 | 0.109465 |
| | DT | 2.2 | 0.83875 | 0.5061 | 0.0 | 0.00025 | 0.03375 |
| Sample #3 [3.7, 0.6160,4.6817] | ANN | 3.2500756 | 0.61871899 | 4.92672224 | 0.449924 | 0.00267 | 0.245022 |
| | GPR | 3.24910094 | 0.61383 | 3.88201961 | 0.45089906 | 0.002205 | 0.79968 |
| | RF | 3.39274603 | 0.61309 | 5.12438397 | 0.30725397 | 0.00294767 | 0.442684 |
| | SVR | 3.153259 | 0.57155 | 2.21052146 | 0.5467414 | 0.04445 | 2.471179 |
| | DT | 3.2 | 0.61604 | 6.1994 | 0.5 | 0.0 | 1.5177 |

## Acknowledgments

The work presented in this contribution has been funded by the University GRANTS COMMISION NEW DELHI, GOVT. OF INDIA under the UGC-NET JRF SCHEME under grant No. 3725/(NET-JULY2018). The funding agency funded to undertake quality research in the chosen topic. However, none of the authors received any salary other than the Fellowship received by Umhara as JRF fellow from the UGC. The authors received no specific funding for this work.

## Author Contributions

**Conceptualization:** Umhara Rasool Khan, Javaid A. Sheikh.

**Formal analysis:** Javaid A. Sheikh.

**Investigation:** Javaid A. Sheikh.

**Methodology:** Umhara Rasool Khan, Shazia Ashraf.

**Validation:** Shazia Ashraf.

**Writing – original draft:** Aqib Junaid.

**Writing – review & editing:** Aqib Junaid, Altaf A. Balkhi.

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
