## [Decision Letter · Decision Letter 0]

12 Oct 2023

PONE-D-23-30368A Machine Learning Driven Computationally Efficient Horse Shoe Shaped Antenna Design for Wearable Internet of Medical ThingsPLOS ONE

Dear Dr. Sheikh,

Thank you for submitting your manuscript to PLOS ONE. After careful consideration, we feel that it has merit but does not fully meet PLOS ONE’s publication criteria as it currently stands. Therefore, we invite you to submit a revised version of the manuscript that addresses the points raised during the review process.

We look forward to receiving your revised manuscript.

Kind regards,

Musa Hussain

Academic Editor

PLOS ONE

Journal Requirements:

   "The funders had no role in study design, data collection and analysis, decision to publish, or preparation of the manuscript"

Reviewers' comments:

Reviewer's Responses to Questions

**Comments to the Author**

1. Is the manuscript technically sound, and do the data support the conclusions?

Reviewer #1: Partly

Reviewer #2: No

2. Has the statistical analysis been performed appropriately and rigorously? 

Reviewer #1: No

Reviewer #2: Yes

3. Have the authors made all data underlying the findings in their manuscript fully available?

Reviewer #1: Yes

Reviewer #2: No

4. Is the manuscript presented in an intelligible fashion and written in standard English?

Reviewer #1: No

Reviewer #2: Yes

5. Review Comments to the Author

Reviewer #1: 1. There are many spelling and grammar mistakes in the text.

2. The use of machine learning is limited to the use of artificial neural networks (ANNs). In this case, it is recommended to use these networks to control the gain, radiation efficiency, and also to reduce the dimensions in terms of wavelength.

3. The quality of many figures is poor.

4. Give each of the Poynting vectors, electric, magnetic fields, and current density distributions at operating frequencies. Full-wave analyzes are not sufficiently presented.

5. Provide 3D SAR reports based on IEEE standards.

6. Justify how this antenna can be used as a wearable antenna. Because in this structure, metals are also used. The dielectric is also not very flexible. Wearability and attachability are two separate issues.

7. The performance of this antenna should be compared with other similar works in the form of a table. Operating frequency, bandwidth, dimensions in terms of wavelength and centimeters, gain, radiation efficiency, etc.

Reviewer #2: The authors presented Machine Learning Driven Computationally Efficient Horse Shoe Shaped Antenna Design for Wearable Internet of Medical Things. The paper is an interesting study, however, before coming toward any decision few things need to be updated and based upon that the decision will be made.

>>>>The quality of the graphs is too low, most of the data is not readable, it is highly recommended to use professional tool and improve the quality.

>>>> Literature work and introducion section need to be updated using latest work as the design of flexible antenna is discussed for body centeric devices in A Conformal Tri-Band Antenna for Flexible Devices and Body-Centric Wireless Communications, 2023 or design of flexible and reconfigurable antenna studied in Electronically reconfigurable and conformal triband antenna for wireless communications systems and portable devices, 2022 or the antennas for IoT applications as discussed in A Series fed Planar Array-based 4-port MIMO Antenna for 5G mmWave IoT Applications, 2023 and Stub loaded Compact Size Tri-Band Antenna for ISM, 5G sub-6-GHz, IoT Applications, 2023.

>>>>The formatting error is found throughout the manuscript, a lot of unnecessary empty spaces need to removed.

>>>>The measurement setup and antenna under test must be shown to verify the results.

>>>>Radiation pattern in both plane must be added and compared with measured results.

>>>>Gain and radiation pattern should be added in the revised manuscript to show the performance of antenna.

>>>>Comparison with state of the art must be added considering latest work not limited to

A low-profile antenna for on-body and off-body applications in the lower and upper ISM and WLAN bands, 2023.

PDMS Based Compact Antenna for 2.45 GHz Application having Wide Band Harmonic Suppresion, 2022

6. PLOS authors have the option to publish the peer review history of their article (what does this mean?). If published, this will include your full peer review and any attached files.

Reviewer #1: No

Reviewer #2: No

---

## [Author Response · Author response to Decision Letter 0]

9 Jan 2024

Original Manuscript ID: PONE-D-23-30368 

Original Article Title: “A Machine Learning Driven Computationally Efficient Horse Shoe Shaped Antenna Design for Wearable Internet of Medical Things"

To: PLOS ONE Editor

Re: Response to reviewers

Dear Editor,

Thank you for allowing a resubmission of our manuscript, with an opportunity to address the reviewers’ comments.

We are uploading (a) our point-by-point response to the comments (below) (response to reviewers), (b) an updated manuscript with yellow highlighting indicating changes (Supplementary Material for Review), and (c) a clean updated manuscript without highlights (Main Manuscript).

Best regards,

Javaid A. Sheikh

Corresponding Author

Reviewer#1, Concern # 1: There are many spelling and grammar mistakes in the text.

Author response: Thank you for your valuable comments on our work

Author action: We updated the manuscript by carrying out necessary corrections. The changes carried out are highlighted with yellow color in the revised manuscript

Reviewer#1, Concern # 2: The use of machine learning is limited to the use of artificial neural networks (ANNs). In this case, it is recommended to use these networks to control the gain, radiation efficiency, and also to reduce the dimensions in terms of wavelength.

Author response: Thank you for your valuable comments on our work

Author action: In this article, we have used ANN, GPR, DT, RF and SVR ML algorithms for predicting the performance of the antenna. The independent variables include relative permittivity of substrate, height above the human body where the antenna is placed, dimensions of the slot incorporated in the ground, radii of the semicircular slots on the patch, length of the ground plane, and dimensions of the feedline. The dependent variables of the models developed include radiation efficiency, Specific Absorption Rate and resonant frequency. These ML models are used to provide an insight to the user regarding the relationship between the input and output variables. So, these ML models are here used to control the output variables like Radiation Efficiency, SAR and frequency of operation which in turn determine the dimensions of the design.

Reviewer#1, Concern # 3: The quality of many figures is poor.

Author response: Thank you for your valuable comments on our work.

Author action: We updated the manuscript by improving the quality of figures. 

Reviewer#1, Concern # 4: Give each of the Poynting vectors, electric, magnetic fields, and current density distributions at operating frequencies. Full-wave analyzes are not sufficiently presented

Author response: Thank you for your valuable comments on our work.

Author action: We updated the manuscript by inserting all the results. The captions of the incorporated figures and the related text is highlighted with yellow color in the revised manuscript

Reviewer#1, Concern # 5: Provide 3D SAR reports based on IEEE standards.

Author response: Thank you for your valuable comments on our work

Author action: We updated the manuscript by incorporating the 3D SAR of the design. The captions of the incorporated figure are highlighted with yellow color in the revised manuscript (Fig. 11)

Reviewer#1, Concern # 6: i. Justify how this antenna can be used as a wearable antenna. Because in this structure, metals are also used. 

ii. The dielectric is also not very flexible. Wearability and attachability are two separate issues. 

Author response: Thank you for your valuable comments on our work 

Author action: 

i. Metals traces that are etched on the substrate work as the radiator and ground, therefore making the circuit complete and making transmission and reception of EM waves possible. The effect of this device on human body is usually evaluated using Specific Absorption rate (SAR) and needs to be less than 2 W/Kg (for 10g of tissue). This has been achieved in the article.

ii. Rogers RT 5880/5870 have extensively been used as substrate materials for antennas in case of wearable/on-body devices as discussed in the following articles. This material is semi-flex in nature and can be used for conformal applications.

[24] W. A. Awan, A. Abbas, S. I. Naqvi, D. H. Elkamchouchi, M. Aslam, and N. Hussain, “A Conformal Tri-Band Antenna for Flexible Devices and Body-Centric Wireless Communications,” Micromachines, vol. 14, no. 10, p. 1842, Sep. 2023, doi: 10.3390/mi14101842. [Online]. Available: http://dx.doi.org/10.3390/mi14101842

A flexible tri-band antenna meant for wearable devices and body-centric wireless communications operating at the key frequency bands is proposed. Rogers RT 5880 is used as a substrate with a mere thickness of 0.254 mm and an overall geometrical dimension of 15 × 20 × 0.254 mm3. This inventive design features a truncated corner monopole accompanied by branched stubs fed by a coplanar waveguide.

[26] A. Arif, M. Zubair, M. Ali, M. U. Khan, and M. Q. Mehmood, ‘‘A compact, low-profile fractal antenna for wearable on-body WBAN applications,’’ IEEE Antennas Wireless Propag. Lett., vol. 18, no. 5, pp. 981–985, May 2019, doi: 10.1109/LAWP.2019.2906829.

A compact and low-profile wearable antenna is presented for on-body wireless body area network (WBAN) applications. The proposed triangular patch antenna is designed using low-cost widely available vinyl polymer-based flexible substrate. The proposed prototype is fabricated using polymer-based flexible substrate Roger RT/duroid 5880 with the thickness of 0.508 mm. The conductive parts of a proposed antenna are etched on copper cladding with an expected conductivity of 5.96 × 107 S/m and thickness of 35 μm. The Rogers substrate has an estimated dielectric constant of 2.20 and a tangent loss of 9 × 10–4 at an operational frequency band.

M. K. Magill, G. A. Conway and W. G. Scanlon, "Circularly Polarized Dual-Mode Wearable Implant Repeater Antenna With Enhanced Into-Body Gain," in IEEE Transactions on Antennas and Propagation, vol. 68, no. 5, pp. 3515-3524, May 2020, doi: 10.1109/TAP.2020.2972335.

A wearable stripline-fed circularly polarized dual-patch antenna structure that exhibits enhanced into-body gain is presented. The antenna is designed for body-surface repeater solutions and it addresses the problem of marginal into-body deep tissue communication links, where power consumption is of the utmost importance and system link efficiency is critical. In this work authors have used Roger RT/duroid 5870 as the patch substrate.

T. Govindan et al., "Design and Analysis of a Conformal MIMO Ingestible Bolus Sensor Antenna for Wireless Capsule Endoscopy for Animal Husbandry," in IEEE Sensors Journal, vol. 23, no. 22, pp. 28150-28158, 15 Nov.15, 2023, doi: 10.1109/JSEN.2023.3323658.

This work portrays the design and analysis of an ingestible capsule multiple-input-multiple-output (MIMO) antenna for wireless capsule endoscopy (WCE) for animal husbandry. The proposed antenna element has a footprint of 0.18λ0×0.107λ0×0.002λ0 , which is transformed into a four-port MIMO antenna and bent into a cylindrical profile of 8.92 mm radius with dimensions of 0.184λ0×0.136λ0×0.002λ0 . The designed MIMO configuration consists of asymmetric extended fork-shaped radiators with U-slotted strips and meandered connected ground planes. The developed MIMO set is housed inside a polylactide (PLA) capsule that mimics a real-time pill camera. The capsule antenna operates at the Industrial, Scientific, and Medical (ISM) (2.45 GHz) and ultra-wideband (UWB) (3.1–10.6 GHz) frequencies. The link budget analysis and diversity metrics of the MIMO antenna are evaluated by embedding it inside the capsule The proposed antenna is developed on the flexible Rogers RT/duroid 5870 substrate (εr = 2.33 and loss tangent = 0.0012) of thickness of 0.26 mm. The two-port antenna is designed with a connected ground plane to reduce mutual coupling. The layout of the antenna and its dimensions are shown in Fig. 2. Figs. 3 and 4 depict the development steps of the antenna and the related reflection coefficient curves..

Reviewer#1, Concern # 7: The performance of this antenna should be compared with other similar works in the form of a table. Operating frequency, bandwidth, dimensions in terms of wavelength and centimeters, gain, radiation efficiency, etc.

Author response: Thank you for your valuable comments on our work

Author action: We updated the manuscript by adding comparison of the state of the art prototypes with the reported design in tabular form at the end of the Discussion section (Table III). The paragraph included is highlighted with yellow color in the revised manuscript.________________________________________

Reviewer#2, Concern # 1: The quality of the graphs is too low, most of the data is not readable, it is highly recommended to use professional tool and improve the quality

Author response: Thank you for your valuable comments on our work

Author action: We updated the manuscript by incorporating the figures with high resolution.

Reviewer#2, Concern # 2: Literature work and introduction section need to be updated using latest work as the design of flexible antenna is discussed for body centric devices in A Conformal Tri-Band Antenna for Flexible Devices and Body-Centric Wireless Communications, 2023 or design of flexible and reconfigurable antenna studied in Electronically reconfigurable and conformal triband antenna for wireless communications systems and portable devices, 2022 or the antennas for IoT applications as discussed in A Series fed Planar Arraybased 4-port MIMO Antenna for 5G mm-Wave IoT Applications, 2023 and Stub loaded Compact Size Tri-Band Antenna for ISM, 5G sub-6-GHz, IoT Applications, 2023. 

Author response: Thank you for your valuable comments on our work

Author action: We updated the introduction section of the manuscript by discussing about some state-of-the-art prototypes that have been recently reported. Furthermore, at the end of the discussion section, comparison of the state of the art prototypes with the reported design is illustrated in tabular form. The paragraph included is highlighted with yellow color in the revised manuscript

Reviewer#2, Concern # 3: The formatting error is found throughout the manuscript, a lot of unnecessary empty spaces need to removed.

Author response: Thank you for your valuable comments on our work

Author action: We updated the article by removing unnecessary spaces in between the paragraphs.

Reviewer#2, Concern # 4: The measurement setup and antenna under test must be shown to verify the results

Author response: Thank you for your valuable comments on our work 

Author action: Image of antenna under measurement has been incorporated in Fig. 4.

Reviewer#2, Concern # 5: Radiation pattern in both plane must be added and compared with measured results. 

Author response: Thank you for your valuable comments on our work

Author action: We updated the manuscript by incorporating all the suggestions. Radiation pattern of antenna when placed on human body in both of the planes has been incorporated in Fig. 9.

Reviewer#2, Concern # 6: Comparison with state of the art must be added considering latest work not limited to

A low-profile antenna for on-body and off-body applications in the lower and upper ISM and WLAN bands, 2023.

PDMS Based Compact Antenna for 2.45 GHz Application having Wide Band Harmonic Suppresion, 2022

Author response: Thank you for your valuable comments on our work

Author action: We updated the manuscript by adding comparison of the state of the art prototypes with the reported design in tabular form at the end of the Discussion section (Table III). The paragraph included is highlighted with yellow color in the revised manuscript.

---

## [Decision Letter · Decision Letter 1]

11 Mar 2024

PONE-D-23-30368R1A Machine Learning Driven Computationally Efficient Horse Shoe Shaped Antenna Design for Wearable Internet of Medical ThingsPLOS ONE

Dear Dr. Sheikh,

Thank you for submitting your manuscript to PLOS ONE. After careful consideration, we feel that it has merit but does not fully meet PLOS ONE’s publication criteria as it currently stands. Therefore, we invite you to submit a revised version of the manuscript that addresses the points raised during the review process. Please consider all the reviewers' comments and address the comments if they are reasonable. Please submit your revised manuscript by Apr 25 2024 11:59PM. If you will need more time than this to complete your revisions, please reply to this message or contact the journal office at plosone@plos.org. Please include the following items when submitting your revised manuscript:A rebuttal letter that responds to each point raised by the academic editor and reviewer(s). You should upload this letter as a separate file labeled 'Response to Reviewers'.A marked-up copy of your manuscript that highlights changes made to the original version. You should upload this as a separate file labeled 'Revised Manuscript with Track Changes'.An unmarked version of your revised paper without tracked changes. You should upload this as a separate file labeled 'Manuscript'.If applicable, we recommend that you deposit your laboratory protocols in protocols.io to enhance the reproducibility of your results. Protocols.io assigns your protocol its own identifier (DOI) so that it can be cited independently in the future. For instructions see: https://journals.plos.org/plosone/s/submission-guidelines#loc-laboratory-protocols. Additionally, PLOS ONE offers an option for publishing peer-reviewed Lab Protocol articles, which describe protocols hosted on protocols.io. Read more information on sharing protocols at https://plos.org/protocols?utm_medium=editorial-email&utm_source=authorletters&utm_campaign=protocols.

We look forward to receiving your revised manuscript.

Kind regards,

Hongzhi Guo, Ph.D.

Academic Editor

PLOS ONE

Journal Requirements:

Reviewers' comments:

Reviewer's Responses to Questions

**Comments to the Author**

1. If the authors have adequately addressed your comments raised in a previous round of review and you feel that this manuscript is now acceptable for publication, you may indicate that here to bypass the “Comments to the Author” section, enter your conflict of interest statement in the “Confidential to Editor” section, and submit your "Accept" recommendation.

Reviewer #1: All comments have been addressed

Reviewer #2: (No Response)

Reviewer #3: (No Response)

Reviewer #4: All comments have been addressed

2. Is the manuscript technically sound, and do the data support the conclusions?

Reviewer #1: Yes

Reviewer #2: Partly

Reviewer #3: Partly

Reviewer #4: Yes

3. Has the statistical analysis been performed appropriately and rigorously? 

Reviewer #1: Yes

Reviewer #2: (No Response)

Reviewer #3: (No Response)

Reviewer #4: Yes

4. Have the authors made all data underlying the findings in their manuscript fully available?

Reviewer #1: Yes

Reviewer #2: Yes

Reviewer #3: Yes

Reviewer #4: Yes

5. Is the manuscript presented in an intelligible fashion and written in standard English?

Reviewer #1: Yes

Reviewer #2: Yes

Reviewer #3: (No Response)

Reviewer #4: Yes

6. Review Comments to the Author

Reviewer #1: This manuscript has been significantly improved. Now this work can be published in the PLOS ONE journal.

Reviewer #2: The researchers have put significant amount of efforts to address the reviwer's comments, however, the literature review section is still weak. The authors are requested to follow the following comments and improve the quality of the manuscript

1) Introduction>paragraph#1: the following statement "There have been serious efforts to address the challenge of optimization under the given constraints for better performance of the system and one such effort is employing Machine learning-based approaches to address the complex and multifaceted challenges while designing the system." can be referred to the 10.1109/TAP.2023.3314097 || 10.1002/adfm.202306249

2) the following statement "In ML techniques, a hidden co-relation between the output and input parameters of the design is determined and accordingly based on this relationship model, future predictions or decisions are made" cab be cited by 10.1007/s11227-021-03898-y || 10.1016/j.nanoen.2023.108387

3) The following passage "Several ML techniques have been discussed in the existing literature for increasing the optimum usage of resources (time, human and computational resources)....." Discuss few of the machine learning techniques and the applications where they are used for instance the usage of machine learning for the hierarchy of the intercity connection of twin cities (10.1016/j.jik.2022.100293) or the posture estimation and tracking based on Long Short Term Memory explained in 10.1016/j.icte.2024.01.002

4) Page 5> Line#4 "It needs to be mentioned here that antennas designed for on-body application is a challenging task as the interaction of RF waves with different tissues of human body make the impedance matching a difficult task" can be referred to 10.3390/polym15173553 || 10.1016/j.cej.2023.146340

5) The following statements

"Also, there is significant reduction in gain of the antenna meant for bio-medical applications owing to the substantial divergence between the dielectric properties of air and tissues of human body, in addition to the change in the frequency of operation for which it is designed"

Therefore, it becomes imperative for the RF engineer to design a wideband, flexible, biocompatible and miniaturized antenna for biomedical applications"

must be referred with propoer reference, here are few suggestions.

10.3390/mi14101842 || 10.3390/s23020709

6) The quality of Fig. 1 is still not improved. Authors are requested to use professional tools like visio or powerpoint.

7) Current distribution grapghs are required to understand the behavior of the antenna radiation.

Reviewer #3: 1. The manuscript contains many grammatical errors. There is significant room for improvement in this regard.

2. The motivation for specifically using horseshoe shaped antenna for on-body applications is not clear.

3. The results presented in Figure 8 are not discussed.

4. In Support Vector Regression, how the optimal values of ε and C are selected?

5. References [17]-[22] are discussed in the manuscript, however, their performance is not compared with the proposed design in Table III.

Reviewer #4: This work presents design of Horeshoe shaped antenna utilizing artificial intelligence, a regression-based Machine learning (ML) techniques. The work is good, but I have few questions.

1. Why there is resonant frequency shift in simulated and measured results in Fig. 5 however antenna works perfectly fine in Fig. 6.

2. Authors must focus on flexibility of antenna as well as reducing SAR of antenna for WBAN applications in their future work.

7. PLOS authors have the option to publish the peer review history of their article (what does this mean?). If published, this will include your full peer review and any attached files.

Reviewer #1: **Yes: **Farzad Khajeh-Khalili

Reviewer #2: **Yes: **Wahaj Abbas Awan

Reviewer #3: No

Reviewer #4: No

---

## [Author Response · Author response to Decision Letter 1]

10 May 2024

Original Manuscript ID: PONE-D-23-30368

Original Article Title: “A Machine Learning Driven Computationally Efficient Horse Shoe Shaped 

Antenna Design for Wearable Internet of Medical Things ”. 

To: PLOS ONE Editor

Re: Response to reviewers

Dear Editor,

Thank you for allowing a resubmission of our manuscript, with an opportunity to address the 

reviewers’ comments.

We are uploading (a) our point-by-point response to the comments (below) (response to reviewers), 

(b) an updated manuscript with yellow highlighting indicating changes (Supplementary Material for 

Review), and (c) a clean updated manuscript without highlights (Main Manuscript).

Best regards,

Javaid A. Skeikh

Corresponding Author

Reviewer#1, Concern # 1: This manuscript has been significantly improved. Now this work can be 

published in the PLOS ONE journal

Author response: Thank you for your valuable comments on our work

Reviewer#2, Concern # 1: Introduction>paragraph#1: the following statement "There have been 

serious efforts to address the challenge of optimization under the given constraints for better 

performance of the system and one such effort is employing Machine learning-based approaches to 

address the complex and multifaceted challenges while designing the system." can be referred to the 

10.1109/TAP.2023.3314097 || 10.1002/adfm.202306249.

Author response: Thank you for your valuable comments on our work

Author action: We have updated the manuscript by incorporating the suggested reference.

Reviewer#2, Concern # 2: the following statement "In ML techniques, a hidden co-relation between 

the output and input parameters of the design is determined and accordingly based on this relationship 

model, future predictions or decisions are made" cab be cited by 10.1007/s11227-021-03898-y || 

10.1016/j.nanoen.2023.108387

Author response: Thank you for your valuable comments on our work.

Author action: We have updated the manuscript by incorporating the suggested reference.

Reviewer#2, Concern # 3: The following passage "Several ML techniques have been discussed in 

the existing literature for increasing the optimum usage of resources (time, human and computational 

resources)....." Discuss few of the machine learning techniques and the applications where they are 

used for instance the usage of machine learning for the hierarchy of the intercity connection of twin 

cities (10.1016/j.jik.2022.100293) or the posture estimation and tracking based on Long Short Term 

Memory explained in 10.1016/j.icte.2024.01.002

Author response: Thank you for your valuable comments on our work.

Author action: We updated the manuscript by inserting all the relevant references.

Reviewer#2, Concern # 4: "It needs to be mentioned here that antennas designed for on-body 

application is a challenging task as the interaction of RF waves with different tissues of human body 

make the impedance matching a difficult task" can be referred to 10.3390/polym15173553 || 

10.1016/j.cej.2023.146340

Author response: Thank you for your valuable comments on our work

Author action: The below mentioned suggested article has no relevance with this statement.

Zou, Y.; Zhong, M.; Li, S.; Qing, Z.; Xing, X.; Gong, G.; Yan, R.; Qin, W.; Shen, J.; Zhang, H.; et al. Flexible Wearable 

Strain Sensors Based on Laser-Induced Graphene for Monitoring Human Physiological Signals. Polymers 2023, 15, 3553. 

https://doi.org/10.3390/polym15173553

Reviewer#2, Concern # 5 "Also, there is significant reduction in gain of the antenna meant for bio-medical applications owing to the substantial divergence between the dielectric properties of air and 

tissues of human body, in addition to the change in the frequency of operation for which it is 

designed"

Therefore, it becomes imperative for the RF engineer to design a wideband, flexible, biocompatible 

and miniaturized antenna for biomedical applications"

must be referred with proper reference, here are few suggestions.

10.3390/mi14101842 || 10.3390/s23020709

Author response: Thank you for your valuable comments on our work

Author action: We updated the manuscript by adding comparison of the state of the art prototypes 

with the reported design in tabular form at the end of the Discussion section (Table III). The paragraph 

included is highlighted with yellow color in the revised manuscript.

Reviewer#2, Concern # 6: The quality of Fig. 1 is still not improved. Authors are requested to use 

professional tools like visio or powerpoint

Author response: Thank you for your valuable comments on our work

Author action: We updated the manuscript by incorporating the figure with high resolution.

Reviewer#2, Concern # 7: Current distribution graphs are required to understand the behavior of the 

antenna radiation.

Author response: Thank you for your valuable comments on our work

Author action: We have updated the manuscript by introducing the current distribution graph along 

with the Electric field distribution in Fig.12 & 13.

Reviewer#3, Concern # 1: The manuscript contains many grammatical errors. There is significant 

room for improvement in this regard.

Author response: Thank you for your valuable comments on our work

Author action: We have updated the article by removing grammatical errors.

Reviewer#3, Concern # 2: The motivation for specifically using horseshoe shaped antenna for on-body applications is not clear.

Author response: Thank you for your valuable comments on our work

Author action: Horseshoe shaped structure is used in design of antenna for modifying the radiation 

pattern

Reviewer#3, Concern #3 : The results presented in Figure 8 are not discussed.

Author response: Thank you for your valuable comments on our work

Author action: We have updated the manuscript and explanation is highlighted in the revised 

manuscript.

Reviewer#3, Concern # 4: In Support Vector Regression, how the optimal values of ε and C are 

selected?

Author response: Thank you for your valuable comments on our work

Author action: The optimal values of ε and C were achieved by performing an exhaustive search 

over the defined hyperparameter grid which provided us with the best combination of 

hyperparameters (SVR model).

Reviewer#3, Concern # 6: References [17]-[22] are discussed in the manuscript, however, their 

performance is not compared with the proposed design in Table III.

Author response: Thank you for your valuable comments on our work

Author action: We updated the manuscript by adding the suggested references in the comparison 

table (Table III). The changes are highlighted with green color in the revised manuscript.

Reviewer#4, Concern # 1: Why there is resonant frequency shift in simulated and measured results 

in Fig. 5 however antenna works perfectly fine in Fig. 6.

Author response: Thank you for your valuable comments on our work

Author action: Figure 5 represents the reflected coefficient for the prototype in free space whereas 

Fig.6 illustrates reflected coefficient in on-body scenario, therefore variation in the resonant 

frequency.

Reviewer#4, Concern # 1: Authors must focus on flexibility of antenna as well as reducing SAR of 

antenna for WBAN applications in their future work.

Author response: Thank you for your valuable comments on our work

Author action: We are currently working on WBAN antennas for specific biomedical applications. 

We will surely work on the suggested concerns raised by you

---

## [Decision Letter · Decision Letter 2]

27 May 2024

A Machine Learning Driven Computationally Efficient Horse Shoe Shaped Antenna Design for Wearable Internet of Medical Things

PONE-D-23-30368R2

Dear Dr. Sheikh,

We’re pleased to inform you that your manuscript has been judged scientifically suitable for publication and will be formally accepted for publication once it meets all outstanding technical requirements.

Kind regards,

Hongzhi Guo, Ph.D.

Academic Editor

PLOS ONE

Additional Editor Comments (optional):

Reviewers' comments:

Reviewer's Responses to Questions

**Comments to the Author**

1. If the authors have adequately addressed your comments raised in a previous round of review and you feel that this manuscript is now acceptable for publication, you may indicate that here to bypass the “Comments to the Author” section, enter your conflict of interest statement in the “Confidential to Editor” section, and submit your "Accept" recommendation.

Reviewer #1: All comments have been addressed

Reviewer #2: All comments have been addressed

Reviewer #3: All comments have been addressed

2. Is the manuscript technically sound, and do the data support the conclusions?

Reviewer #1: Yes

Reviewer #2: Yes

Reviewer #3: Partly

3. Has the statistical analysis been performed appropriately and rigorously? 

Reviewer #1: Yes

Reviewer #2: Yes

Reviewer #3: Yes

4. Have the authors made all data underlying the findings in their manuscript fully available?

Reviewer #1: Yes

Reviewer #2: Yes

Reviewer #3: Yes

5. Is the manuscript presented in an intelligible fashion and written in standard English?

Reviewer #1: Yes

Reviewer #2: Yes

Reviewer #3: Yes

6. Review Comments to the Author

Reviewer #1: The present manuscript is suitable for publication.

Previous concerns have been addressed. It is only recommended that the authors, together with the editors, re-review the manuscript text in terms of grammar and phrasing.

Reviewer #2: The authors address all the comments carefully. The manuscript is technically sound and is recommended for publication.

Reviewer #3: The authors have addressed all the technical concerns raised by the reviewer. However, there are still several punctuation and grammatical errors. It is recommended that professional English writing services may be sought before final acceptance of the paper.

7. PLOS authors have the option to publish the peer review history of their article (what does this mean?). If published, this will include your full peer review and any attached files.

Reviewer #1: **Yes: **Dr. Farzad Khajeh-Khalili

Reviewer #2: No

Reviewer #3: No

---

## [Editor Report · Acceptance letter]

28 Aug 2024

PONE-D-23-30368R2 

PLOS ONE

Dear Dr. Sheikh, 

I'm pleased to inform you that your manuscript has been deemed suitable for publication in PLOS ONE. Congratulations! Your manuscript is now being handed over to our production team.

Kind regards, 

on behalf of

Dr. Hongzhi Guo 

Academic Editor

PLOS ONE